# SwiftSage: A Generative Agent with
# Fast and Slow Thinking for Complex Interactive Tasks

**Bill Yuchen Lin** [1]   **Yicheng Fu** [2]   **Karina Yang** [3]   **Prithviraj Ammanabrolu** [1]   **Faeze Brahman** [1]   **Shiyu Huang** [4]
**Chandra Bhagavatula** [1]   **Yejin Choi** [1,5]   **Xiang Ren** [1,3]

## Abstract

We introduce SWIFTSAGE, a novel agent framework inspired by the dual-process theory of human cognition, designed to excel in action planning for complex interactive reasoning tasks. SWIFTSAGE integrates the strengths of behavior cloning and prompting large language models (LLMs) to enhance task completion performance. The framework comprises two primary modules: the SWIFT module, representing fast and intuitive thinking, and the SAGE module, emulating deliberate thought processes. The SWIFT module is a small encoder-decoder LM fine-tuned on the oracle agent's action trajectories, while the SAGE module employs LLMs such as GPT-4 for subgoal planning and grounding. We develop a heuristic method to harmoniously integrate the two modules, resulting in a more efficient and robust problem-solving process. In 30 tasks from the ScienceWorld benchmark, SWIFTSAGE significantly outperforms other methods such as Say-Can, ReAct, and Reflexion, demonstrating its effectiveness in solving complex real-world tasks.

## 1. Introduction

The advancement of artificial general intelligence is largely dependent on the development of agents that are proficient in complex interactive reasoning tasks. These agents should be capable of exhibiting problem-solving abilities akin to humans within dynamic, open-world environments (Reed et al., 2022; Bubeck et al., 2023). For example, the Science-World benchmark (Wang et al., 2022) features a task where an agent must determine the electrical conductivity of an unknown object. In a simulated environment, the agent must navigate to appropriate rooms, locate and acquire essen-

tial items, such as batteries and light bulbs, build a circuit, perform an experiment, and interpret the results. Tackling such a complex interactive task demands agents to exhibit long-horizon planning, long-term memorization, subgoal decomposition, spatial reasoning, exception handling, and commonsense knowledge capabilities (Wang et al., 2023b).

There are three primary approaches to developing agents capable of addressing complex interactive reasoning tasks: (1) deep reinforcement learning (RL), (2) behavior cloning (BC) (Torabi et al., 2018) through sequence-to-sequence (seq2seq) learning (Sutskever et al., 2014), and (3) prompting large language models (LLMs) (Brown et al., 2020). In addition to conventional RL methods such as DRRN (He et al., 2016), interactive reasoning can be framed as a seq2seq task, where the input text serves as the current state description and the output text corresponds to the subsequent action (Chen et al., 2021; Ammanabrolu et al., 2021). By leveraging numerous gold trajectories generated by oracle agents, it becomes feasible to fine-tune Transformer models (Vaswani et al., 2017), like T5 (Raffel et al., 2020), to effectively imitate the behavior of these oracle agents. Recent studies have also demonstrated that generative agents based on prompting LLMs, such as GPT-4, can produce reasonable plans and actions (Lin et al., 2022; Huang et al., 2022; Song et al., 2022).

Although the aforementioned methods exhibit remarkable performance in relatively simple tasks, their ability to generalize to more complex and demanding tasks is limited. Both RL-based and seq2seq-based BC approaches effectively acquire knowledge from the environment through large-scale interactions and learn general action patterns from oracle agents. However, they face difficulties in decomposing tasks into subgoals, maintaining long-term memory, generalizing to unseen tasks, and handling exceptions. In contrast, instruction-tuned LLMs (Ouyang et al., 2022) demonstrate the ability to generate reasonable high-level plans for complex tasks and adapt their outputs based on human feedback. Yet, grounding their outputs to executable actions in the environment remains a challenge. These procedures also lack the capability to efficiently handle environment-specific exceptions that prevent agents from adhering to

Website: https://yuchenlin.xyz/swiftsage/. Affiliations: [1]Allen Institute for AI [2]Tsinghua University [3]University of Southern California [4]4Paradigm Inc. [5]University of Washington. Correspondence to: Bill Yuchen Lin <yuchenl@allenai.org>.

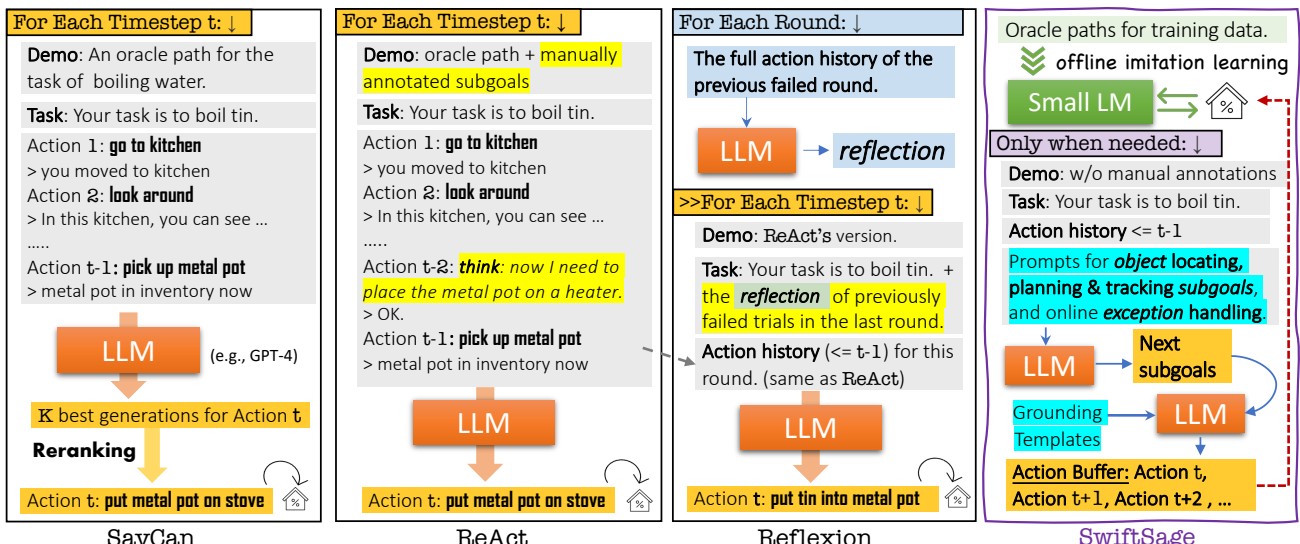

*Figure 1.* **Comparing methods of prompting LLMs to build agents for interactive tasks.**

the LLM's plans. Additionally, previous methods such as SAYCAN (Ahn et al., 2022), REACT (Yao et al., 2022) and REFLEXION (Shinn et al., 2023) require a new inference with LLMs for each time step, making them considerably costly and inefficient (see Figure 1).

Inspired by the dual process theory (Wason & Evans, 1974; Kahneman, 2011), we propose a novel framework that enables agents to closely emulate how humans solve complex, open-world tasks. The dual-process theory posits that human cognition is composed of two distinct systems: System 1, characterized by rapid, intuitive, and automatic thinking; and System 2, which entails methodical, analytical, and deliberate thought processes. System 1 is reminiscent of seq2seq methods, which learn through imitation of oracle agents and primarily operate utilizing shallow action patterns. Conversely, System 2 bears resemblance to LLMs that excel in applying commonsense knowledge, engaging in step-by-step reasoning, devising subgoal strategies, and exercising self-reflection. Thus, our proposed method, SWIFT-SAGE, is designed to enable both fast and slow thinking in complex interactive reasoning tasks. It effectively integrates the strengths of behavior cloning (representing System 1) and prompting LLMs (emulating System 2), resulting in significant enhancements in task completion performance and efficiency.

Specifically, SWIFTSAGE consists of two primary modules: the SWIFT module and the SAGE module. The SWIFT module is a small encoder-decoder LM, fine-tuned on a T5-large (770m) checkpoint using the searched oracle trajectories of training tasks. It encodes short-term memory components, such as previous actions, observations, visited locations, as well as the current environment state. Then, it decodes

the next individual action. This module simulates the fast, intuitive thinking characteristic of System 1. The SAGE module, representing the deliberate thinking of System 2, utilizes LLMs, such as GPT-4, and is structured around two prompting stages: planning and grounding. In the planning stage, we prompt LLMs to locate necessary items, plan and track subgoals, as well as detect and fix potential exceptions and mistakes. In the grounding stage, we focus on utilizing LLMs to transform the output subgoals derived from the planning stage into a sequence of actions by demonstrating potential action templates. Unlike prior methods, where LLMs only generate the next immediate action, our procedures engage in longer-term action planning. To harmoniously integrate the SWIFT and SAGE modules, we developed a heuristic algorithm that determines when to (de)activate the SAGE module and how to combine the outputs effectively with an action buffer mechanism.

In a comprehensive evaluation on 30 task types from the ScienceWorld benchmark, SWIFTSAGE significantly outperforms other methods, achieving a state-of-the-art average score of 84.7. In comparison, SAYCAN scores 33.8, REACT obtains 36.4, and REFLEXION reaches 45.3. Moreover, SWIFTSAGE is more cost-effective and efficient, requiring much fewer tokens per action for LLM inference than previous methods. This considerable performance advantage highlights the effectiveness and efficiency of the SWIFT-SAGE framework in addressing complex interactive tasks.

## 2. Background and Related Work

We first provide a formal introduction to complex interactive reasoning tasks, with a particular emphasis on the

ScienceWorld benchmark. Subsequently, we present a summary of existing methods that can be adapted for interactive reasoning, which also serve as baseline methods in our experiments. Lastly, we discuss the dual-process theory, which serves as a significant inspiration for our work.

## 2.1. Complex Interactive Reasoning

We define interactive reasoning as the problems where agents are tasked with accomplishing a goal within an interactive environment, typically simulated by engines such as AI2Thor (Kolve et al., 2017) and TextWorld (Côté et al., 2018). Our focus lies on the textual environment of Science-World (Wang et al., 2022) and the complex interactive tasks it supports. Simple interactive tasks, like those created in ALFWorld (Shridhar et al., 2021) and TWC (Murugesan et al., 2021), primarily involve searching for and placing objects as well as performing basic actions within a single location. Many of these simple tasks have been almost solved by recent works.

In contrast, tasks in ScienceWorld exhibit greater complexity, characterized by more challenging task planning and a significantly larger action space (encompassing 10 locations, 200 types of objects with varying states, and 25 types of actions). Furthermore, agents may encounter random, unforeseen obstacles, such as broken stoves or missing soil, which hinder the execution of planned actions. As a result, agents must adapt and re-plan accordingly, for example, by seeking alternative heat sources or using a shovel on the outside ground to get soil. These challenges demand that agents possess skills in long-horizon planning, long-term memory, subgoal decomposition, exception handling, and commonsense knowledge—capabilities that are not explicitly required for simple interactive tasks.

## 2.2. Reinforcement Learning & Imitation Learning

**DRRN.** Interactive tasks can naturally be framed as partially-observable Markov decision processes (POMDPs), enabling the application of RL-based methods. Deep Reinforced Relevance Network (DRRN) (He et al., 2016) is a standard baseline method to learn agents within text environment. It aims to learn representations of observations and actions separately and train a policy network to select actions from candidates based on feedback from the simulated environment. **CALM** (Yao et al., 2020) is a reranking-based method that combines DRRN with a causal language model (LM) fine-tuned with oracle transcripts. In essence, the causal LM captures task-specific and environment-specific knowledge through imitation learning, and the DRRN learns to rerank the predictions from the LM.

The **KG-A2C** (Ammanabrolu & Hausknecht, 2020) method uses an OpenIE technique (Angeli et al., 2015) to represent environment states with graph structures and dynamically update these graphs. These graphs guide policy networks by constraining the combinations of action templates and objects. This method has been shown to be effective in other domains such as for multimodal embodied agents (Nottingham et al., 2023).

**Behavior cloning for offline imitation learning.** Behavior cloning is an imitation learning method that trains a seq2seq Transformer offline with action transcripts of similar training tasks generated by oracle agents (Torabi et al., 2018; Ammanabrolu et al., 2021). During training, it uses the previous action, observation at time step $t - 1$, and the current observation as input and learns to output the next action. The Text Decision Transformer (**TDT**) is a textual variant of the Decision Transformer (Chen et al., 2021), which also employs behavior cloning and uses the same data. The primary innovation of TDT is the introduction of reward-to-go as part of the inputs, enabling the model to learn predicting actions that maximize future expected rewards.

## 2.3. Prompting LLMs for Action Planning.

Language models (LLMs) such as GPT-4 have shown promise for action planning in interactive tasks (Lin et al., 2022; Huang et al., 2022; Song et al., 2022; Wang et al., 2023a). In this paper, we adapt three prominent methods to complex interactive reasoning tasks in ScienceWorld: SAY-CAN (Ahn et al., 2022), REACT (Yao et al., 2022), and REFLEXION (Shinn et al., 2023).

**SAYCAN** (Ahn et al., 2022) is a straightforward agent that integrates an LLM with a value function of underlying policies regarding grounding affordances (i.e., the feasibility of an action in the environment). We need to provide the history and current environment as textual inputs to LLMs for generating a ranked list of action candidates. This action list is then reranked based on a value function.

**REACT** (Yao et al., 2022) presents a virtual 'think' action, enabling LLMs to generate *subgoals* during action planning. This approach requires human annotators to supply examples of correct subgoals for each task type, employing few-shot in-context learning to teach LLMs *when* and *how* to 'think' in order to plan subsequent subgoals, in addition to providing complete action trajectories.

**REFLEXION** (Shinn et al., 2023), a recent work building on REACT, proposes a multi-round approach enabling LLMs to use the history of previously failed rounds to refine their planning for the next round. This self-reflection mechanism helps LLMs improve after each failed attempt. However, this may not be practical in real-world applications for many tasks, as actions in failed trials can be irrecoverable.

All three methods require a new LLM inference at each time

step to predict the next immediate action, resulting in inefficient and costly agents. REACT and REFLEXION require human annotations of correct subgoals for each unseen task type. Moreover, it is difficult to generalize REFLEXION to real-world situations where trial-and-error approaches can be infeasible for embodied tasks.

## 2.4. Dual-Process Theory

The dual-process theory (Wason & Evans, 1974; Kahneman, 2011) is a cognitive psychological framework proposing the existence of a fast and a slow thinking systems in the human brain. This influential theory has found widespread applications across various fields, highlighting the critical role of both systems in shaping human cognition (Anthony et al., 2017; Chen et al., 2019; Ganapini et al., 2021; Miech et al., 2021). By integrating the complementary strengths of both systems, agents can effectively and efficiently handle diverse challenges in real-world scenarios. Inspired by this, we aim to construct a generative agent that utilizes a small seq2seq LM as System 1 for associative reasoning via behavior cloning while developing System 2 for analytical reasoning by prompting LLMs.

## 3. SWIFTSAGE: A Generative Agent with Fast and Slow Thinking

### 3.1. Problem Formulation

**Environment and tasks.** We focus on complex interactive reasoning tasks situated in virtual textual environments such as ScienceWorld (Wang et al., 2022). ScienceWorld provides an optimal setting for developing and evaluating agents in *complex* tasks, comprising 30 distinct task types covering 10 topics in science experiments. It features 10 locations, including an art studio, workshop, kitchen, living room, bedroom, bathroom, foundry, greenhouse, outdoor area, and a connecting hallway. The environment includes 200+ object types with multiple states (e.g., open, activated) and supports 25 action templates, resulting in an intractable search space. The simulator can generate numerous variations of each task type, providing a rich training ground. In each variation, the agent and environment initialization, such as the locations and states of objects, will differ. A plethora of training variations encompassing all task types are available for training agents. Additionally, it provides a handcrafted oracle agent to search for successful transcripts with minimal actions for offline learning.

Evaluation is done on a set of testing variations with unseen combinations of required objects and situations, thus substantially different from the training variations. For example, a training variation may involve boiling water, while a testing variation could require boiling tin. Therefore, it is crucial to ensure the agent's compositional generalization

ability for effectively handling real-world scenarios.

**Interactions.** Given a task variation, an agent is provided with the task description $D$ and the initial environment state ($t = 0$). The task description $D$ is a text specifying a high-level goal, e.g., "*Your task is to test if an unknown substance A is electronically conductive.*" At each time step $t$, the agent generates an action $A_t$ based on a set of supported action templates (e.g., `pick up X`, `use X on Y`). $A_0$ is always "`look around`" for showing initial environment information. Upon receiving an action from the agent, the environment produces feedback in four dimensions:

- **Observation** $O_t$ provides direct feedback on the action $A_t$ regarding its effects on the environment or the information queried. For example, an $A_t$ of "`use thermometer on the substance in metal pot`" may result in an $O_t$ like "*The temperature is 80F.*"

- **Environment** $E_t$ represents the current room in which the agent is situated and provides details about all visible objects. Object visibility is based on container states, e.g., objects within a closed fridge are not included in $E_t$ until the agent performs an action like "`open fridge`."

- **Inventory** $I_t$ lists objects picked up by the agent, which is particularly useful when agents collect items from different locations to complete the task.

- **Score** $S_t$ represents the agent's cumulative score ranging from 0 to 100. When a required intermediate state is achieved, the score increases with a positive reward.

### 3.2. SWIFT: The Module for Intuitive and Associative Thinking via Imitation Learning

Imitation learning is used to construct an agent that learns to mimic oracle agents in various training scenarios through seq2seq learning. Previous methods, such as TDT (Wang et al., 2022), mainly employ one-hop history as input context and learn to output the subsequent action $A_t$ (Wang et al., 2022). However, these methods exhibit limitations due to their *restricted context* of action history and harmful biases arising from *data imbalance*. To address these issues, we introduce our SWIFT module, depicted in Figure 2.

**Representation for longer history.** We expand the conventional one-hop BC to multi-hop by incorporating a sliding window of observations and rewards for the $K = 10$ recent actions. Additionally, we include a special field for visited rooms (without duplication). This approach aims to provide agents with a longer context and prevent unnecessary room navigation. The input format is as follows: "`Task:` $D$; `Time:` $t - 1$; `Score:` $S_{t-1}$; `Action history:` [$A_{t-i}$ ($+R_{t-i}$) $\rightarrow O_{t-i}$ ] /* $i$ loops from $K$ to 1*/; `Current room:` $E_{t-1}$; `Inventory:` $I_{t-1}$; `Visited rooms:` $\{E_1^*, \ldots, E_{t-1}^*\}$". Here, $R_t = S_t - S_{t-1}$ represents the

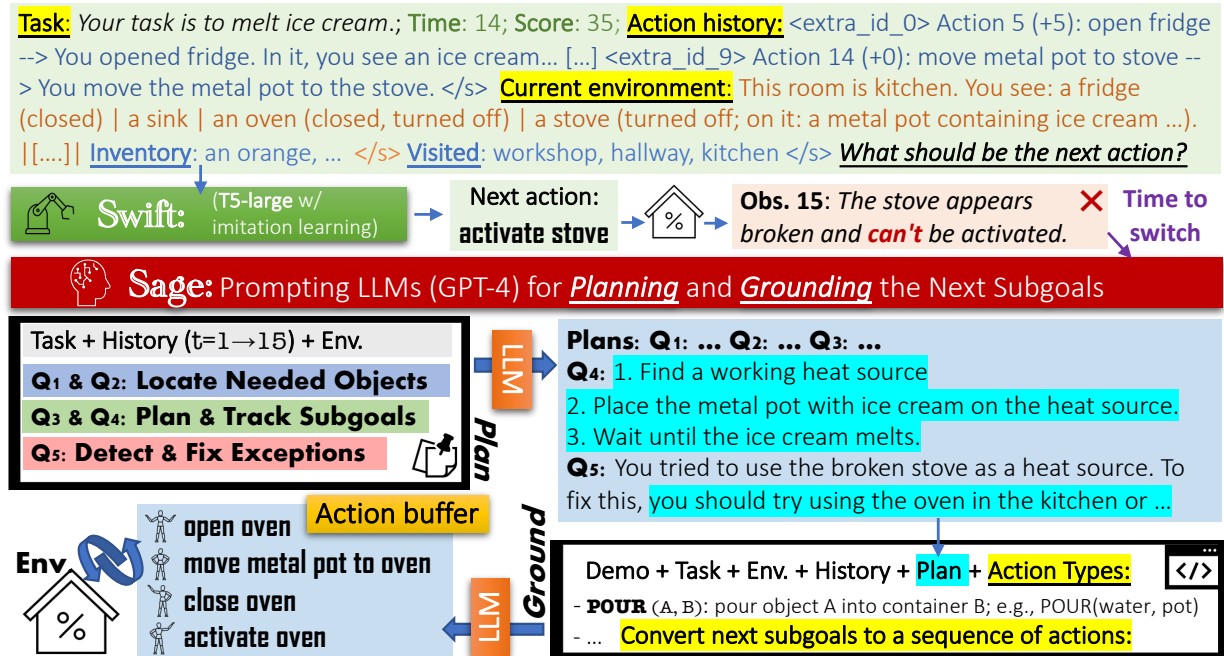

*Figure 2.* **An example of how SWIFTSAGE works with fast and slow thinking.** The SWIFT module is offline trained via imitation learning with a small LM such as T5-large (770m). When it is necessary, for example, encountering an exception, we switch to the SAGE module that prompts LLMs (e.g., GPT-4) for planning and grounding the next subgoals, resulting in an action buffer.

reward at $t$, and $E_t^*$ is the location name at $t$.

**Balanced imitation learning.** To avoid bias caused by data imbalance for seq2seq learning, we down-sampled specific types of tasks and actions to achieve a more balanced final dataset for training. We used the T5-large with 770 million parameter and instruction-following ability (Chung et al., 2022), creating an efficient agent that we named SWIFT. Our empirical results show that the SWIFT module performs much better than TDT (11 billion) despite being 15x smaller in size.

The SWIFT module exhibits greater accuracy during initial time steps, enabling it to attain higher scores in the early stages of a complex task. However, it often fails to generalize to unseen situations. The module also has a tendency to repeat meaningless actions when its learned plans yield exceptions from the environment (e.g., the broken stove in Figure 2). This is partly due to the nature of imitation learning, which prioritizes emulating the observable actions of oracle agents rather than their intrinsic planning abilities. Besides, since the oracle trajectories contain only the shortest, correct actions, it is thus also challenging for the SWIFT to learn how to fix mistaken actions.

### 3.3. SAGE: The Module for Deliberate and Analytical Thinking via Prompting LLMs

While the SWIFT module acquires surface knowledge about the environment and task types through imitation learning, it lacks two key abilities essential for complex interactive reasoning: 1) generalizable planning and tracking of subgoals, and 2) robust handling of exceptions. Prior research has shown that LLMs outperform smaller LMs in these abilities. They can perform step-by-step reasoning to devise concrete plans for tasks and self-refine their outcomes. However, the performance of prior methods remains unsatisfactory in complex interactive tasks such as those in ScienceWorld.

We introduce a novel two-stage approach, named SWIFT-SAGE. This method initially acquires higher-level recommendations from LLMs during the planning stage, followed by their translation into specific action sequences in the grounding stage. By decoupling the planning and grounding processes, SWIFTSAGE effectively generates a series of actions for completing the planned subgoals.

**Planning stage.** In this stage, we leverage LLMs to plan based on the current state. Specifically, we prompt LLMs with a single prompt that includes a summarized version of the task description and action history, and asks the following five key questions.

Before posing the five planning-related questions, we con-

dense the entire action history ($A_{<t}$ and $O_{<t}$), and the current environment information $E_{t-1}$. **Q1** and **Q2** pertain to objects, as acquiring all necessary objects serves as the foundation for effective task planning. By addressing these questions, we ensure that LLMs develop a comprehensive understanding of the current environment. **Q3** prompts LLMs to engage in step-by-step planning by decomposing the task into a series of subgoals. **Q4** acts as a follow-up question, allowing the agent to monitor its progress based on the action history and determine completed subgoals, subsequently focusing on the remaining tasks. Lastly, **Q5** is employed to identify and address potential exceptions. These questions can be further tailored with additional environment-specific hints, thereby enhancing their adaptability.

> ▶ **Q1**(`locate objects`): "*To complete the task, which objects do I need to collect? Please list them and their possible locations one by one.*"
>
> ▶ **Q2**(`track objects`): "*Are there any objects that have not been collected yet?*"
>
> ▶ **Q3**(`plan subgoals`): "*To complete the task most efficiently, what are the important subgoals to achieve? Please list the subgoals one by one.*"
>
> ▶ **Q4**(`track progress`): "*Considering these subgoals, what have I already completed? And which subgoal should I focus on right now?*"
>
> ▶ **Q5**(`handle exceptions`): "*Have I made any mistakes that might prevent me from efficiently completing the next subgoal? If any , how should I fix them?*"

To improve the structure of the LLMs' outputs and facilitate parsing, we incorporate additional instructions in the prompt. By utilizing a *single* input to obtain answers to all five questions in one output, rather than engaging in multiple rounds of interactive prompting, our approach is more efficient and cost-effective than the iterative prompting methods.

**Q4** and **Q5** are of primary importance, while **Q1−Q3** serve as auxiliary guidance for the LLMs. If the action history indicates a mistaken action or an unachievable previous subgoal, the response to **Q5** refines the answer to **Q4** through *self-reflection on the fly*. This approach differs from the REFLEXION agent, which only prompts reflective questions at the end of a failed trial, allowing agents to improve their planning in subsequent attempts. In contrast, our method detects exceptions and errors each time the agent plans for the next subgoals, enabling earlier correction of the agent's behavior.

**Grounding stage.** While the answers to **Q1−Q5** provide valuable guidance for agents, they are not directly executable. Converting plans into valid actions that can be accepted by the environment remains a challenge. Previ-

ous methods using LLMs over-generate candidates, and they rely on reranking or filtering based on the action space to select the next action. However, this is inefficient and inaccurate for complex tasks with vast action spaces. Additionally, these methods generate a single action at a time, which can be both costly and ineffective for long-horizon tasks.

To tackle these issues, we first present supported action types using a formal style accompanied by remarks. For instance, the action type "`pour X into Y`" is introduced as "`POUR(X, Y)`: *pour object X into container Y; e.g., pour red paint into wood cup*". We then incorporate the LLM's outputs from the planning stage as part of the input for the grounding stage. Furthermore, we provide the recent action history of the past 10 time steps as context. Finally, we prompt LLMs to concentrate on the next subgoal and convert it into a *list* of actions (rather than a single action) to accomplish the next subgoal. Our formatting instructions enable the straightforward splitting and conversion of output actions from LLMs in the grounding stage back to their original action representations. We denote this list of actions generated by LLMs as the *action buffer*: $B = \{\hat{A}_t, \hat{A}_{t+1}, \dots\}$.

### 3.4. Integration of Fast and Slow Thinking

Having described the SWIFT and SAGE modules, we now address the question of how to merge both modules and effectively integrate fast and slow thinking within the SWIFTSAGE agent. We establish a heuristic algorithm to control the activation and deactivation of the two modules.

Initially, we employ the SWIFT module due to its superior intuitive reasoning capabilities, which facilitate accurate associations between the task description and the environment during the first few actions. We switch from SWIFT mode to SAGE when any of the following conditions are met:

> 1) There are five consecutive time steps with zero reward ($\sum_{i=t-5}^{t-1} R_i = 0$).
>
> 2) The SWIFT's prediction for the next action ($A'_t$) is invalid in the current environment.
>
> 3) $A'_t$ can result in a critical decision, such as giving the final answer for the experiment result.
>
> 4) The observation of $A'_t$ suggests that an exception is encountered.

Upon activating the SAGE module, we execute the two-stage prompting process and generate an action buffer. We attempt to execute each predicted action and revert to the SWIFT module when the buffer is empty. This enables a seamless integration of both modules, providing an efficient and robust reasoning process for the SWIFTSAGE agent.

# 4. Evaluation

## 4.1. Evaluation Setup

To evaluate the effectiveness of SWIFTSAGE and other baseline methods in complex interactive reasoning tasks, we use the ScienceWorld benchmark. In Section 2.1 and Section 3.1, we introduce the benchmark and problem formulation. Each task type is categorized as 'short' (S), 'medium' (M), or 'long' (L) based on the average length of the oracle truth trajectories. However, the length of the task does not necessarily indicate its level of difficulty as some tasks may require additional commonsense knowledge. Further evaluation details are provided in the appendix.

## 4.2. Baseline Agents

In addition to the baseline methods evaluated in the ScienceWorld paper, such as DRRN, CALM, KG-A2C, and TDT, we incorporate three LLM-based prompting techniques: SAYCAN, REACT, and REFLEXION, as detailed in Section 2.3 and Figure 1. This subsection presents the implementation details for adapting these methods to build ScienceWorld agents.

SAYCAN necessitates a value function from the environment for reranking purposes. We employ Sentence-BERT (Reimers & Gurevych, 2019) to rank all valid actions (generated by ScienceWorld's APIs) based on their similarity to the top 5 generations for $A_t$ from SAYCAN. We implemented REACT and REFLEXION in a similar manner. Adhering to their released code, we utilized the best single generation and determined the valid action with the minimal edit distance, if required. Both REACT and REFLEXION necessitate subgoal annotations for teaching LLMs to plan with virtual 'think' actions. We annotated such truth subgoals by translating ScienceWorld's APIs into natural language, which was also employed by the oracle agents. For all agents, we incorporated the complete trajectories of one or two training variations from the same task type for in-context learning. Our primary experiments were conducted using OpenAI's GPT-4; however, other LLMs can be readily substituted as required.

## 4.3. Results and analysis.

**Main Results** Table 1 compares the performance of various agents across 30 types of tasks. Detailed descriptions of each task type can be found in the ScienceWorld paper (Wang et al., 2022) and our appendix. It is evident that LLM-based methods outperform conventional agents due to their superior generalization ability, albeit at a higher deployment cost. The behavior cloning model TDT (Wang et al., 2022; Chen et al., 2021) (11b) performs on par with DRRN (He et al., 2016), but with greater efficiency in learning and inference. In contrast, our SWIFT-only agent (770m)

achieves an overall performance of 49.22, which we attribute to its balanced training data and the use of a sliding window for longer action histories.

REACT demonstrates a noticeable improvement over SAY-CAN for short and medium tasks, owing to its subgoal annotations for in-context learning and the inclusion of 'think' actions. REFLEXION surpasses REACT in shorter tasks; however, comparing REFLEXION with other agents is not entirely fair. REFLEXION can run up to four rounds, while the others are limited to one round. This discrepancy is particularly unfair for tasks involving multiple-choice scenarios. Nevertheless, we include REFLEXION's results to analyze the potential of such methods.

**Exception handling.** Consider the example in Figure 2, where the stove is broken, presenting an exception. Agents like DRRN and TDT often resort to repeating meaningless action sequences (e.g., continuously attempting to activate the stove or moving between rooms aimlessly). Although the SWIFT module, when used independently, improves upon this due to its larger context window from imitation learning, it still struggles to address exceptions robustly. ReAct and Reflexion occasionally utilize the 'think' action or reflections to redirect agents towards alternative solutions, but the generated actions rarely achieve the new subgoals if they are not grounded. In contrast, the plan-and-ground prompts in our SAGE module handle exceptions more effectively.

**Cost-effectiveness.** Despite SAGE invoking LLMs APIs twice for inference, its overall cost remains lower, as the result is a *sequence* of actions typically containing about 5 actions. In comparison, SAYCAN and REACT require **1,855.84** and **1,971.03** *tokens per action* (tpa) respectively, while REFLEXION necessitates **2,983.46** tpa. SWIFTSAGE, on the other hand, only uses **757.07** tpa. Given its superior performance, SWIFTSAGE proves more cost-effective than other LLM-based methods. This efficiency is primarily attributed to invoking LLMs only when needed (courtesy of our strong SWIFT module) and the action buffer mechanism.

**Efficiency.** To thoroughly examine the efficiency of agents across all task types, we use Figure 3 to visualize the average trajectories of the first three testing variations for each task involving SWIFTSAGE, REACT, and the oracle agent. We arrange the tasks based on their average lengths of oracle trajectories (*Len* in Table 1). We observe that oracle trajectories consistently achieve perfect scores, yet SWIFT-SAGE can reach similar scores more efficiently. This is particularly evident in longer tasks (the bottom two rows), although SWIFTSAGE does not achieve a perfect score for a few tasks (e.g., 9-2 and 1-3). Interestingly, we find that REACT performs competitively in shorter tasks (e.g., 4-2

| Task Type | *Len | DRRN | KGA2C | CALM | TDT | SayCan | ReAct | Reflexion | SwiftSage |
|---|---|---|---|---|---|---|---|---|---|
| **1**-1 (L) | 107.7 | 3.52 | 0.0 | 0.0 | 0.71 | 33.06 | 3.52 | 4.22 | 97.04 |
| 1-2 (L) | 78.6 | 3.52 | 0.0 | 0.0 | 0.44 | 10.39 | 13.70 | 10.61 | 87.04 |
| 1-3 (L) | 88.9 | 0.0 | 4.0 | 0.0 | 3.88 | 3.88 | 7.78 | 7.78 | 72.78 |
| 1-4 (L) | 75.2 | 0.0 | 0.0 | 0.0 | 0.55 | 0.37 | 9.88 | 0.92 | 100.0 |
| **2**-1 (M) | 21.4 | 6.56 | 6.0 | 1.0 | 6.16 | 26.37 | 7.19 | 5.92 | 99.17 |
| 2-2 (M) | 35.2 | 5.50 | 11.0 | 1.0 | 6.43 | 8.03 | 6.10 | 28.59 | 88.17 |
| 2-3 (L) | 65.0 | 6.0 | 4.0 | 1.0 | 19.87 | 17.41 | 22.37 | 22.37 | 95.73 |
| **3**-1 (S) | 13.6 | 12.0 | 7.0 | 5.0 | 40.55 | 52.14 | 56.0 | 100.0 | 88.67 |
| 3-2 (M) | 20.8 | 9.0 | 4.0 | 7.0 | 14.26 | 22.50 | 54.33 | 17.45 | 55.33 |
| 3-3 (M) | 25.6 | 9.05 | 4.0 | 2.0 | 10.16 | 99.56 | 76.19 | 72.54 | 71.90 |
| 3-4 (M) | 29.0 | 9.52 | 4.0 | 2.0 | 21.65 | 47.76 | 88.81 | 70.22 | 77.86 |
| **4**-1 (S) | 14.6 | 15.0 | 18.0 | 10.0 | 41.93 | 22.87 | 26.67 | 64.93 | 100.0 |
| 4-2 (S) | 8.8 | 45.0 | 44.0 | 54.0 | 55.76 | 58.18 | 80.0 | 87.27 | 100.0 |
| 4-3 (S) | 12.6 | 21.67 | 16.0 | 10.0 | 27.82 | 20.87 | 53.33 | 16.42 | 91.67 |
| 4-4 (S) | 14.6 | 19.17 | 15.0 | 8.0 | 47.15 | 31.43 | 27.50 | 100.0 | 100.0 |
| **5**-1 (L) | 69.5 | 8.0 | 6.0 | 2.0 | 6.89 | 9.92 | 9.06 | 7.33 | 74.59 |
| 5-2 (L) | 79.6 | 14.29 | 11.0 | 4.0 | 11.86 | 13.93 | 18.57 | 13.0 | 93.93 |
| **6**-1 (M) | 33.6 | 15.77 | 17.0 | 3.0 | 15.10 | 47.81 | 51.04 | 70.35 | 49.40 |
| 6-2 (S) | 15.1 | 26.67 | 19.0 | 6.0 | 15.70 | 39.26 | 58.89 | 70.67 | 100.0 |
| 6-3 (M) | 23.0 | 10.37 | 4.0 | 3.0 | 5.25 | 19.72 | 40.74 | 15.77 | 91.48 |
| **7**-1 (S) | 7.0 | 50.0 | 43.0 | 6.0 | 30.0 | 80.0 | 60.0 | 100.0 | 95.0 |
| 7-2 (S) | 7.0 | 50.0 | 32.0 | 10.0 | 8.43 | 67.50 | 67.50 | 84.37 | 85.0 |
| 7-3 (S) | 8.0 | 33.33 | 23.0 | 4.0 | 8.34 | 50.0 | 50.0 | 83.0 | 93.33 |
| **8**-1 (M) | 40.0 | 21.0 | 5.0 | 4.0 | 3.86 | 20.91 | 27.67 | 2.58 | 89.0 |
| 8-2 (S) | 16.3 | 8.0 | 10.0 | 0.0 | 8.0 | 16.0 | 8.0 | 8.0 | 68.50 |
| **9**-1 (L) | 97.0 | 10.0 | 4.0 | 0.0 | 2.53 | 21.94 | 40.50 | 50.63 | 75.0 |
| 9-2 (L) | 84.9 | 10.0 | 4.0 | 3.0 | 14.66 | 32.26 | 44.0 | 100.0 | 70.0 |
| 9-3 (L) | 123.1 | 10.0 | 4.0 | 2.0 | 9.12 | 13.67 | 41.0 | 70.62 | 60.0 |
| **10**-1 (L) | 130.1 | 16.80 | 11.0 | 2.0 | 1.51 | 67.53 | 25.70 | 50.90 | 92.30 |
| 10-2 (L) | 132.1 | 17.0 | 11.0 | 2.0 | 1.29 | 59.45 | 16.80 | 23.69 | 77.60 |
| Short | *11.76* | 28.08 | 22.70 | 11.30 | 28.37 | 43.83 | 48.79 | 71.47 | 92.22 |
| Medium | *28.58* | 10.85 | 6.88 | 2.88 | 10.36 | 36.58 | 44.01 | 35.43 | 77.79 |
| Long | *94.30* | 8.26 | 4.92 | 1.33 | 6.11 | 23.65 | 21.07 | 30.17 | 83.0 |
| **Overall** | *49.26* | **15.56** | **11.37** | **5.07** | **14.66** | **33.82** | **36.43** | **45.34** | **84.68** |

*Table 1.* **Results on the ScienceWorld benchmark.** *Len* is the average length of the oracle agent's trajectories. We report performance on three groups of *Len* (short, medium, long). The last four methods use GPT-4 as the base LLM for prompting.

and 3-4), but most trajectories plateau at an intermediate score and fail to reach 100.

**More analysis.** Due to page limit, we have to provide further details and analysis in the appendix, including more detailed analysis on cost-effectiveness and efficiency, additional case studies and abalation studies, sensitivity to LLM choices, and an the evaluation of the SWIFT-only agent.

## 5. Conclusion

**Contributions.** We present SWIFTSAGE, a novel generative agent for complex interactive reasoning tasks, inspired by the dual-process theory of human cognition. The framework comprises two modules: SWIFT, responsible for fast thinking, and SAGE, dedicated to slow thinking. The SWIFT module is a smaller LM that mimics oracle agents' behavior, while the SAGE module focuses on prompting LLMs for subgoal planning and action sequence grounding. Through extensive experiments on 30 distinct tasks within the ScienceWorld benchmark, SWIFTSAGE outperforms baseline agents, achieving state-of-the-art performance, increased efficiency, and reduced cost.

**Implications.** The success of SWIFTSAGE highlights the potential for collaborative frameworks combining smaller LMs and LLMs in complex reasoning tasks. Smaller LMs can be trained more easily to recognize task-specific and environment-specific patterns, fostering effective in-distribution generalization. On the other hand, LLMs

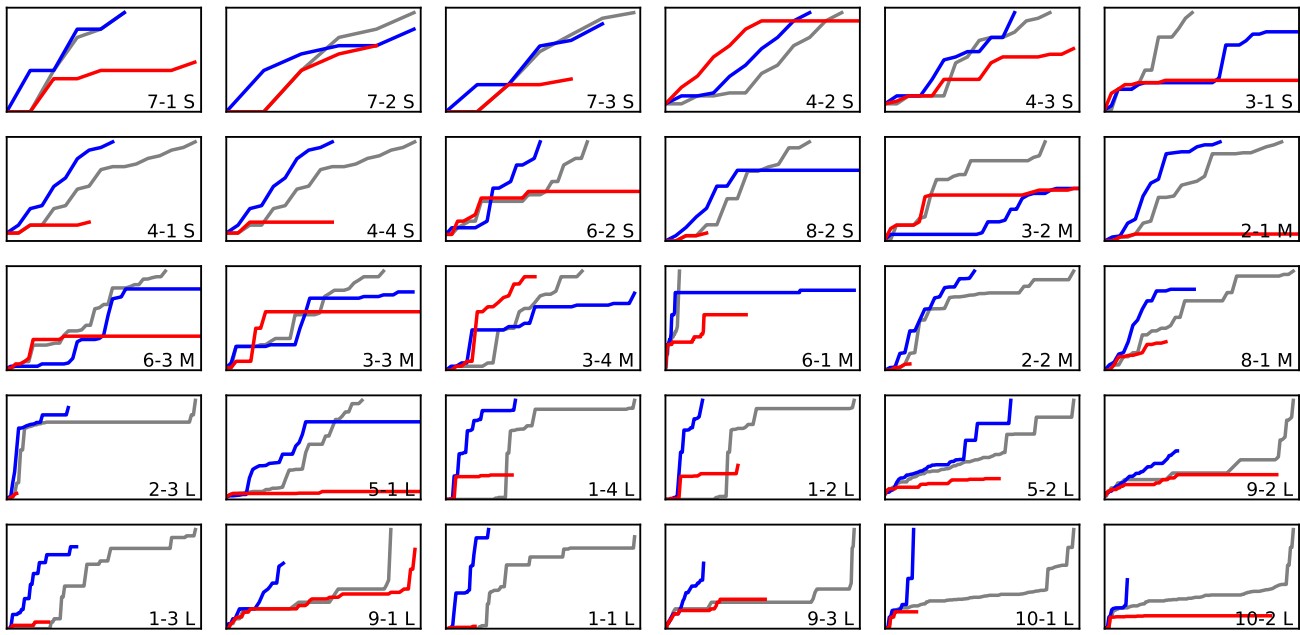

*Figure 3.* **Visualizing trajectories of SWIFTSAGE, REACT and** ORACLE. $X$: time steps $(0 \to T)$; $Y$: scores $(0 \to 100)$. Each figure displays the merged trajectories of testing variations by an agent. Task IDs are bottom-right, and the ordering is based on *\*Len*.

demonstrate remarkable zero-shot generalization abilities and deliberate thinking, though grounding their outputs in real-world environments remains challenging. We posit that dual-process agents, harnessing the strengths of both approaches, constitute a crucial step towards addressing complex interactive reasoning tasks and building general AI agents. We can regard SWIFTSAGE as a method of utilizing LLMs as controllers or planners for decomposing complex tasks and leveraging APIs/tools (Lu et al., 2023; Ge et al., 2023; Shen et al., 2023; Schick et al., 2023).

**Limitations.** Our work has been evaluated solely within a *textual* simulator, ScienceWorld, which supports a limited set of actions and tasks compared to real-world situations. Also, we did not implement any safeguards to prevent agents from engaging in potentially hazardous actions that could occur in the real world, such as picking up substances from a blast furnace. We argue that one important future direction is to develop a true open-ended environment, allowing agents to interact with a much wider variety of actions and objects to better emulate real-world scenarios. Besides, the use of LLMs in SAGE may present scalability challenges, as LLMs require significant computational resources and may not be feasible in some settings. Future research should explore the generalizability of SWIFTSAGE to other domains and the potential for more lightweight approaches to slow thinking.

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

## Appendix

## A. Limitations

Our work has been evaluated solely within a *textual* simulator, ScienceWorld, which supports a limited set of actions and tasks compared to real-world situations. Also, we did not implement any safeguards to prevent agents from engaging in potentially hazardous actions that could occur in the real world, such as picking up substances from a blast furnace. We argue that one important future direction is to develop a true open-ended environment, allowing agents to interact with a much wider variety of actions and objects to better emulate real-world scenarios. Besides, the use of LLMs in SAGE may present scalability challenges, as LLMs require significant computational resources and may not be feasible in some settings. Future research should explore the generalizability of SWIFTSAGE to other domains and the potential for more lightweight approaches to slow thinking.

## B. Dataset Statistics

Table 2 presents the details of all 30 types of tasks in the ScienceWorld benchmark. To improve the training of SWIFT, we down-sampled Task 9-x, 10-x, and 3-3 from the original full dataset, as their large sizes resulted in a significant data imbalance. Additionally, we down-sampled less informative actions, such as 'close door to kitchen,' to produce a more effective dataset for imitation learning.

**Evaluation.**  To save time while evaluating the numerous tasks and agents, we only used the first 10 variations for tasks with more than 10 test variations. This resulted in a total of 270 variations for fair and cost-effective comparisons among all agents. Some agents may receive a negative score from the engine and be unable to proceed any further due to their final action violating task requirements and being irrecoverable. In such cases, we used their last non-negative scores for evaluation.

## C. Implementation Details

### C.1. Training Details of SWIFT

We utilized `flan-t5-large` (770m) as the base model and fine-tuned it using the seq2seq action-prediction data (62k) as previously described. A learning rate of `1e-4` and batch size of `128` were employed for training 500 steps, selected based on dev loss. Although we experimented with larger sizes of flan-t5 models, we observed only marginal improvements at a much higher training cost. We believe this is because the language used to describe the environment and actions covers a small vocabulary, and the language complexity does not warrant the use of more parameters.

### C.2. Prompting in SAGE

In Section 3.3, we provided an overview of the two-stage prompting framework: planning and grounding. In this section, we delve into further details of each stage.

**Memory augmentation.**  Since the agent can only perceive objects in its current environment location, objects from previously visited locations are not displayed unless a prior 'look around' action has been executed. To augment memory for LLMs during planning and grounding, we also present the objects observed in previously visited locations. Additionally, we include the agent's location during each action in the action history, e.g., "pick up metal pot [location: kitchen]," to facilitate spatial reasoning for LLMs.

**Connecting the two stages.**  We conveniently reuse the LLM output from the first stage (i.e., answers to Q1-Q5) as part of the input for the second stage. Our experiments involve using answers to all questions in the grounding stage. However, one can opt to use only answers to Q4 and Q5 to reduce computational costs. Our small-scale ablation study indicates that incorporating answers to Q1-Q3 in the grounding stage proves beneficial, yielding a gain of about 2 points for short tasks on average.

**Grounding with action templates.**  We previously introduced an action template, 'POUR(object A, object B)', in Figure 2. Here, we present several additional templates to further illustrate the concept:

```
TELEPORT(room) : directly go to a room such as
TELEPORT(kitchen)

PICK(object) : pick up an object and put it into
your inventory

OPEN(object) : open an object to search or put
things in it, e.g., OPEN(freezer).

ACTIVATE(object) : activate / turn on an object
such as sink or stove, so that you can use it.

DEACTIVATE(object) : deactivate / turn off the
object

EXAMINE(object) : look at an object carefully. For
example, EXAMINE(light bulb).

MOVE(object, place) : move/place the object to
a place
```

It should be noted that despite explicitly instructing the LLM to *only* utilize permitted action types, it may occasionally generate actions of disallowed types that cannot be parsed. These invalid actions will be disregarded in the action buffer, and if necessary, the system will revert to the SWIFT mode.

| Task Type | Topic | Name | *Lens | #Vars: Train | Dev | Test | # Actions |
|-----------|-------|------|-------|--------------|-----|------|-----------|
| 1-1 | Matter | Changes of State (Boiling) | 107.7 | 14 | 7 | 9 | 694 |
| 1-2 | Matter | Changes of State (Melting) | 78.6 | 14 | 7 | 9 | 427 |
| 1-3 | Matter | Changes of State (Freezing) | 88.9 | 14 | 7 | 9 | 469 |
| 1-4 | Matter | Changes of State (Any) | 75.2 | 14 | 7 | 9 | 344 |
| 2-1 | Measurement | Use Thermometer | 21.4 | 270 | 10 | 10 | 4278 |
| 2-2 | Measurement | Measuring Boiling Point (known) | 35.2 | 218 | 10 | 10 | 6511 |
| 2-3 | Measurement | Measuring Boiling Point (unknown) | 65 | 150 | 10 | 10 | 9768 |
| 3-1 | Electricity | Create a circuit | 13.6 | 10 | 5 | 5 | 94 |
| 3-2 | Electricity | Renewable vs Non-renewable Energy | 20.8 | 10 | 5 | 5 | 169 |
| 3-3 | Electricity | Test Conductivity (known) | 25.6 | 48 | 10 | 10 | 1341 |
| 3-4 | Electricity | Test Conductivity (unknown) | 29 | 300 | 10 | 10 | 6974 |
| 4-1 | Classification | Find a living thing | 14.6 | 150 | 10 | 10 | 1606 |
| 4-2 | Classification | Find a non-living thing | 8.8 | 150 | 10 | 10 | 756 |
| 4-3 | Classification | Find a plant | 12.6 | 150 | 10 | 10 | 1458 |
| 4-4 | Classification | Find an animal | 14.6 | 150 | 10 | 10 | 1606 |
| 5-1 | Biology | Grow a plant | 69.5 | 62 | 10 | 10 | 3675 |
| 5-2 | Biology | Grow a fruit | 79.6 | 62 | 10 | 10 | 4283 |
| 6-1 | Chemistry | Mixing (generic) | 33.6 | 16 | 8 | 8 | 347 |
| 6-2 | Chemistry | Mixing paints (secondary colours) | 15.1 | 18 | 9 | 9 | 224 |
| 6-3 | Chemistry | Mixing paints (tertiary colours) | 23 | 18 | 9 | 9 | 350 |
| 7-1 | Biology | Identify longest-lived animal | 7 | 62 | 10 | 10 | 298 |
| 7-2 | Biology | Identify shortest-lived animal | 7 | 62 | 10 | 10 | 298 |
| 7-3 | Biology | Identify longest-then-shortest-lived animal | 8 | 62 | 10 | 10 | 360 |
| 8-1 | Biology | Identify life stages (plant) | 40 | 6 | 3 | 5 | 165 |
| 8-2 | Biology | Identify life stages (animal) | 16.3 | 4 | 2 | 4 | 31 |
| 9-1 | Forces | Inclined Planes (determine angle) | 97 | 24 | 10 | 10 | 2733 |
| 9-2 | Forces | Friction (known surfaces) | 84.9 | 26 | 10 | 9 | 3644 |
| 9-3 | Forces | Friction (unknown surfaces) | 123.1 | 23 | 10 | 10 | 3284 |
| 10-1 | Biology | Mendelian Genetics (known plants) | 130.1 | 26 | 10 | 10 | 3043 |
| 10-2 | Biology | Mendelian Genetics (unknown plants) | 132.1 | 24 | 10 | 10 | 2853 |
| **Short** $(0 < *\text{Len} \leq 20)$ | | | 11.76 | 81.80 | 8.6 | 8.80 | 673.10 |
| **Medium** $(20 < *\text{Len} \leq 50)$ | | | 28.58 | 110.75 | 8.13 | 8.38 | 2516.88 |
| **Long** $(*\text{Len} > 50)$ | | | 94.30 | 37.75 | 9.00 | 9.58 | 2934.75 |
| **Overall** (avg) | | | 49.26 | 71.90 | 8.63 | 9 | 2069.43 |
| **Overall** (sum) | | | N/A | 2,157 | 259 | 270 | 62,083 |

*Table 2.* **The statistics of ScienceWorld benchmark.** *Len* is the average length of the oracle agent's trajectories. We show the number of our down-sampled variations in each split. The last column is the number of data points forr action-prediction seq2seq task in training SWIFT.

### C.3. Action Buffer

In Section 3.4, we presented four conditions for activating the SAGE module. To detect critical decisions (Condition 3), we primarily focus on the 'focus on' actions, as many tasks in ScienceWorld necessitate agents to concentrate on the correct substances and objects in the proper sequence. A single incorrect 'focus on' action can terminate the entire run. Thus, we restrict the SWIFT module from performing such actions if SAGE has not yet been activated.

For identifying exceptions (Condition 4), we examine phrases like "No known action can match," "... cannot/doesn't...," and so on. When processing an action buffer,

we attempt to execute each action sequentially. If two consecutive actions are invalid or cause exceptions, we halt and revert to SWIFT.

## D. Additional Results and Analysis

### D.1. Sensitivity to LLMs: GPT-3.5-turbo vs GPT-4

Besides the empirical results in Table 1, we also evaluate performance using GPT-3.5-turbo instead of GPT-4, which is considerably larger and more expensive. Other methods exhibit a significant performance decline, for instance, Re-Act's score drops from 36.43 to 19.76, which is close to non-LLM methods and even lower than the vanilla method,

SayCan. In contrast, SWIFTSAGE maintains a respectable performance of 62.22, indicating better robustness.

As discussed in Sec. 5 (*limitations*), we plan to utilize other open-source LLMs, such as Alpaca, and investigate distilling the planning ability from closed-source LLMs to open-source and smaller LMs. Nevertheless, a practical challenge arises due to the current open-source LLMs having more restrictive length limits for inputs and outputs.

### D.2. Efficiency Analysis

Figure 4 illustrates that most of SWIFTSAGE's curves are situated near the top-left corner, indicating that SWIFTSAGE attains higher scores than oracle agents at a faster rate. Although ReAct is competitive with our method for shorter tasks, its trajectories typically plateau at intermediate scores and do not reach 100. While the ORACLE agent consistently achieves a perfect score (100.0), its efficiency, particularly in longer tasks, is often outperformed by SWIFTSAGE.

### D.3. Cost-effectiveness

Table 4 presents a comprehensive analysis of the cost-effectiveness of LLM-based methods. We examine two specific metrics: tokens per action (tpa) and scores per action (spa) for SayCan, ReAct, Reflexion, and SWIFTSAGE across all tasks. Despite SAGE invoking LLM APIs twice for inference, its overall cost remains lower, as the result is a *sequence* of actions typically containing about 5 actions. In contrast, SAYCAN and REACT require **1,855.84** and **1,971.03** *tokens per action* (tpa) respectively, while REFLEXION necessitates **2,983.46** tpa. SWIFTSAGE, however, only uses **757.07** tpa. Given its superior performance, SWIFTSAGE proves to be more cost-effective than other LLM-based methods. This efficiency primarily stems from invoking LLMs only when necessary (thanks to our robust SWIFT module) and the action buffer mechanism.

Interestingly, we observe that SWIFTSAGE has an even lower tpa for long tasks compared to its tpa in medium and short tasks. Upon further investigation, we attribute this finding to longer action buffers and the SWIFT module being more frequently effective. Additionally, regarding scores per action (spa), we discover that our SWIFTSAGE is more cost-effective by utilizing fewer tokens and achieving higher scores.

| Task Type | Swift-Only | SayCan$_{ChatGPT}$ | ReAct$_{ChatGPT}$ | Reflexion$_{ChatGPT}$ | SwiftSage$_{ChatGPT}$ |
|:---:|:---:|:---:|:---:|:---:|:---:|
| 1-1 | 15.0 | 0.0 | 0.4 | 0.7 | 58.0 |
| 1-2 | 24.4 | 0.0 | 8.5 | 0.0 | 58.5 |
| 1-3 | 32.2 | 0.0 | 1.1 | 0.0 | 38.5 |
| 1-4 | 57.4 | 0.0 | 0.6 | 0.0 | 62.5 |
| 2-1 | 9.4 | 1.7 | 4.1 | 2.2 | 47.9 |
| 2-2 | 6.7 | 14.1 | 7.2 | 2.5 | 53.3 |
| 2-3 | 5.7 | 93.7 | 19.6 | 14.7 | 48.6 |
| 3-1 | 70.0 | 19.3 | 16.7 | 20.0 | 72.7 |
| 3-2 | 48.3 | 8.7 | 9.7 | 8.6 | 50.3 |
| 3-3 | 59.5 | 22.0 | 55.5 | 6.4 | 66.9 |
| 3-4 | 69.0 | 36.4 | 36.4 | 30.1 | 78.1 |
| 4-1 | 100.0 | 11.7 | 17.5 | 46.5 | 100.0 |
| 4-2 | 100.0 | 76.0 | 73.3 | 68.2 | 97.5 |
| 4-3 | 94.4 | 11.4 | 20.0 | 19.9 | 58.3 |
| 4-4 | 100.0 | 9.5 | 15.8 | 41.0 | 100.0 |
| 5-1 | 13.4 | 11.3 | 11.1 | 5.8 | 57.5 |
| 5-2 | 44.6 | 75.0 | 18.8 | 47.6 | 50.9 |
| 6-1 | 26.2 | 13.5 | 35.0 | 22.4 | 43.2 |
| 6-2 | 53.3 | 25.0 | 20.0 | 10.0 | 63.3 |
| 6-3 | 11.1 | 58.4 | 16.7 | 40.0 | 27.4 |
| 7-1 | 83.3 | 75.0 | 37.5 | 75.0 | 75.0 |
| 7-2 | 100.0 | 100.0 | 50.0 | 75.0 | 60.0 |
| 7-3 | 77.8 | 31.7 | 31.7 | 28.1 | 68.3 |
| 8-1 | 33.0 | 5.6 | 4.2 | 2.8 | 75.6 |
| 8-2 | 8.0 | 12.8 | 7.0 | 8.2 | 33.0 |
| 9-1 | 73.3 | 38.0 | 28.5 | 100.0 | 54.0 |
| 9-2 | 73.3 | 4.2 | 10.0 | 17.5 | 63.3 |
| 9-3 | 53.3 | 0.0 | 0.0 | 1.7 | 77.0 |
| 10-1 | 17.0 | 1.3 | 24.5 | 1.3 | 76.0 |
| 10-2 | 17.0 | 0.3 | 11.7 | 6.0 | 51.1 |
| Short | 78.68 | 37.24 | 28.95 | 39.19 | 72.81 |
| Medium | 32.90 | 20.06 | 21.09 | 14.37 | 55.34 |
| Long | 35.55 | 18.66 | 11.23 | 16.27 | 57.99 |
| **Overall** | 49.22 | 25.22 | 19.76 | 23.40 | 62.22 |

*Table 3.* **Additional results on the ScienceWorld benchmark.** Different from Table 1, we use `gpt-3.5-turbo` instead of `gpt-4` as the LLM for evaluating SayCan, ReAct, Relfexion, and our SWIFTSAGE. We also present the results of using SWIFT module only.

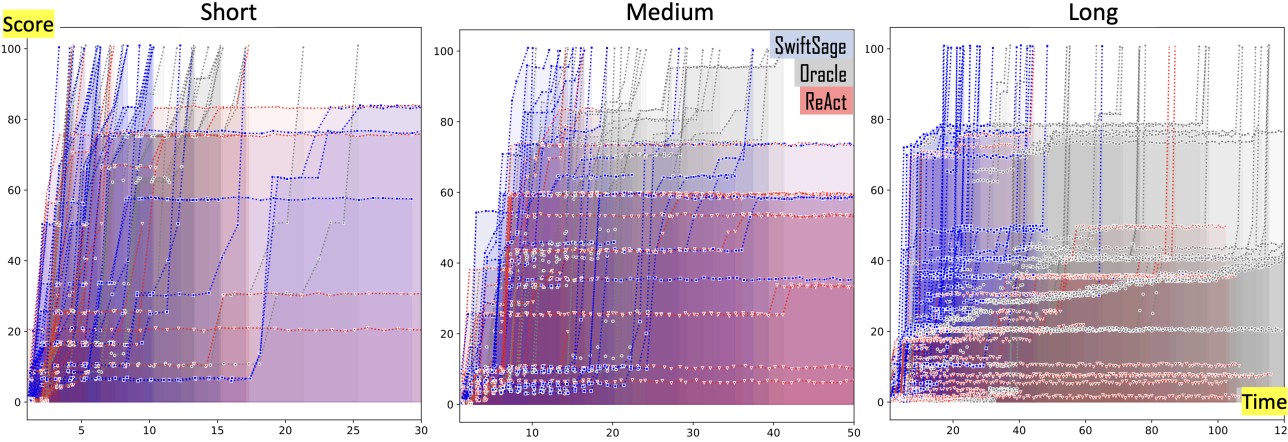

*Figure 4.* **An overview of visualizing trajectories of** **SWIFTSAGE**, **REACT** **and** ORACLE. $X$: time steps $(0 \rightarrow T)$; $Y$: scores $(0 \rightarrow 100)$. Similar to Figure 3, each curve is a single trajectory by an agent in performing a task variation. A more efficient agent will achieve higher scores in a shorter time, resulting in curves positioned near the top-left corner.

| Task Type | Average number of tokens per action (tpa) | | | | Average scores per action (spa) | | | |
|---|---|---|---|---|---|---|---|---|
| | SayCan | ReAct | Reflexion | SwiftSage | SayCan | ReAct | Reflexion | SwiftSage |
| 1-1 | 1944.94 | 1503.60 | 2632.97 | 528.17 | 0.10 | 0.05 | 0.00 | 1.49 |
| 1-2 | 1125.76 | 1339.39 | 3066.70 | 545.34 | 0.08 | 0.13 | 0.01 | 1.64 |
| 1-3 | 1034.33 | 1268.23 | 3307.30 | 550.17 | 0.04 | 0.13 | 0.00 | 0.71 |
| 1-4 | 1295.03 | 1251.45 | 2439.34 | 754.05 | 0.00 | 0.09 | 0.00 | 1.69 |
| 2-1 | 1188.46 | 1545.03 | 1988.59 | 494.52 | 0.13 | 0.06 | 0.00 | 3.01 |
| 2-2 | 1862.11 | 1181.88 | 1596.03 | 394.29 | 0.03 | 0.25 | 0.15 | 2.32 |
| 2-3 | 939.17 | 1358.33 | 1753.17 | 574.05 | 0.73 | 0.93 | 0.12 | 1.71 |
| 3-1 | 1713.64 | 1846.91 | 2677.89 | 807.62 | 0.49 | 0.44 | 0.04 | 0.49 |
| 3-2 | 1785.01 | 1754.14 | 2337.02 | 823.28 | 0.21 | 0.22 | 0.01 | 0.31 |
| 3-3 | 1762.13 | 2441.79 | 2262.39 | 220.80 | 0.53 | 0.18 | 0.10 | 0.84 |
| 3-4 | 1698.85 | 1195.59 | 2859.30 | 287.25 | 0.18 | 1.93 | 0.10 | 1.13 |
| 4-1 | 411.08 | 579.70 | 1053.57 | 309.14 | 1.91 | 1.16 | 0.34 | 4.76 |
| 4-2 | 1332.83 | 1098.69 | 1250.37 | 298.48 | 0.20 | 0.69 | 0.47 | 4.76 |
| 4-3 | 1155.99 | 1314.74 | 2966.82 | 406.17 | 0.08 | 0.39 | 0.02 | 3.82 |
| 4-4 | 1126.67 | 591.15 | 1003.18 | 309.71 | 0.24 | 1.02 | 0.79 | 4.76 |
| 5-1 | 2323.43 | 2620.66 | 5091.49 | 168.95 | 0.02 | 0.02 | 0.00 | 0.22 |
| 5-2 | 2646.50 | 2575.11 | 5864.93 | 536.56 | 0.03 | 0.13 | 0.00 | 0.75 |
| 6-1 | 1454.65 | 1802.62 | 2344.90 | 1388.89 | 0.28 | 0.25 | 0.04 | 0.37 |
| 6-2 | 2413.99 | 2763.66 | 4342.07 | 402.50 | 0.18 | 0.14 | 0.02 | 3.33 |
| 6-3 | 1371.50 | 2860.68 | 4551.96 | 6361.79 | 0.09 | 0.10 | 0.00 | 0.59 |
| 7-1 | 376.50 | 495.83 | 813.08 | 768.63 | 5.71 | 3.33 | 0.77 | 11.88 |
| 7-2 | 424.53 | 478.09 | 1180.58 | 772.00 | 2.11 | 6.14 | 0.36 | 10.63 |
| 7-3 | 424.73 | 564.69 | 1175.35 | 609.73 | 4.55 | 3.85 | 0.24 | 8.48 |
| 8-1 | 1505.39 | 1155.71 | 2466.59 | 249.38 | 0.21 | 0.79 | 0.00 | 2.23 |
| 8-2 | 3189.80 | 741.71 | 2886.09 | 2479.00 | 0.12 | 0.47 | 0.02 | 4.03 |
| 9-1 | 2066.06 | 2642.79 | 2652.56 | 307.30 | 0.16 | 0.14 | 0.08 | 1.06 |
| 9-2 | 2517.48 | 3031.95 | 3606.60 | 314.19 | 0.11 | 0.14 | 0.05 | 0.66 |
| 9-3 | 7002.72 | 7507.00 | 7785.29 | 366.06 | 0.04 | 0.18 | 0.04 | 0.65 |
| 10-1 | 3612.33 | 4218.44 | 4822.97 | 466.21 | 0.32 | 0.36 | 0.13 | 1.78 |
| 10-2 | 3969.62 | 5401.37 | 6724.81 | 218.00 | 0.52 | 0.10 | 0.01 | 1.69 |
| Short | 1256.98 | 1047.52 | 1934.90 | 716.30 | 1.56 | 1.76 | 0.31 | 5.69 |
| Medium | 1578.51 | 1742.18 | 2550.85 | 1277.52 | 0.21 | 0.47 | 0.05 | 1.35 |
| Long | 2539.78 | 2893.19 | 4145.68 | 444.09 | 0.18 | 0.20 | 0.04 | 1.17 |
| Overall | 1855.84 | 1971.03 | 2983.46 | 757.07 | 0.65 | 0.79 | 0.13 | 2.73 |

*Table 4.* **Cost-effectiveness analysis for LLM-based methods.**

