# OpenReview forum: "SwiftSage: A Generative Agent with Fast and Slow Thinking for Complex Interactive Tasks"
_ICML.cc/2023/Workshop/ILHF — ILHF Workshop ICML 2023_

### Official Review · Reviewer_HiNv · 2023-06-04
**An llm-based agent for scienceworld**

**Rating:** 6
**Confidence:** 4

**Review:**

The paper proposes SwiftSage, and algorithm composed of two language models that is used for planning. The smaller and faster language model outputs the next immediate action based on all currently available information. The larger LLM is queried less frequently to produce semantic information about how to solve the task, plan subgoals, and produce a sequence of actions that solves the task. The smaller LM is trained on in-domain data while the larger LM is only pretrained. The paper shows that this outperforms other ways of using pretrained LMs as well as other RL and IL methods.

It appears that the proposed method is the only evaluated method to leverage both pretraining and in-domain data, which is a significant drawback of the paper.

Overall, the paper proposes an interesting algorithm for combining pretraining and finetuning which is an interesting contribution.

---

### Official Review · Reviewer_BcQk · 2023-06-13
**Two-level decision making based on LMs**

**Rating:** 6
**Confidence:** 4

**Review:**

This paper presents a framework for decision-making that utilizes two types of LMs -- a small one finetuned on offline data via imitation learning and a large one without finetuning. The small LM (swift module) directly outputs actions and the large LM (sage module) generates subgoals and ground them to the environment via carefully crafted prompts. The proposed method uses a few heuristics to decide when to use the swift module and when to use the sage module.

Strengths:
1. An interesting framework that uses LMs for policy learning and planning.
2. The comparison of different ways to use LMs for decision-making is very informative.
3. The performance looks promising.

Weaknesses:
1. My main concern is that the method heavily relies on hand-crafted heuristics. It is unclear how general the method is.
2. The connection to the workshop theme is unclear. Is there any human feedback here?

---

### Decision · Program_Chairs · 2023-06-20

Accept